# Study on 1550 nm Human Eye-Safe High-Power Tunnel Junction Quantum Well Laser

**DOI:** 10.3390/mi15081042

**Published:** 2024-08-17

**Authors:** Qi Wu, Dongxin Xu, Xuehuan Ma, Zaijin Li, Yi Qu, Zhongliang Qiao, Guojun Liu, Zhibin Zhao, Lina Zeng, Hao Chen, Lin Li, Lianhe Li

**Affiliations:** 1College of Physics and Electronic Engineering, Hainan Normal University, Haikou 571158, China; wuqihainan@163.com (Q.W.); jilinchangchun@yeah.net (D.X.); 17385729163@163.com (X.M.); lizaijin@hainu.edu.cn (Z.L.); qzhl060910@hainnu.edu.cn (Z.Q.); gjliu626@126.com (G.L.); 060111@hainnu.edu.cn (Z.Z.); zenglina@hainnu.edu.cn (L.Z.); 15948713468@163.com (H.C.); lin.li@hainnu.edu.cn (L.L.); 2Hainan Provincial Key Laboratory of Laser Technology and Optoelectronic Functional Materials, Innovation Center of Hainan Academician Team, Haikou 571158, China; 3Hainan International Joint Research Center for Semiconductor Lasers, Hainan Normal University, Haikou 571158, China; lilianhhnnu@126.com

**Keywords:** 1550 nm LD, human eye-safe band, quantum well laser, tunnel junction

## Abstract

Falling within the safe bands for human eyes, 1550 nm semiconductor lasers have a wide range of applications in the fields of LIDAR, fast-ranging long-distance optical communication, and gas sensing. The 1550 nm human eye-safe high-power tunnel junction quantum well laser developed in this paper uses three quantum well structures connected by two tunnel junctions as the active region; photolithography and etching were performed to form two trenches perpendicular to the direction of the epitaxial layer growth with a depth exceeding the tunnel junction, and the trenches were finally filled with oxides to reduce the extension current. Finally, a 1550 nm InGaAlAs quantum well laser with a pulsed peak power of 31 W at 30 A (10 KHz, 100 ns) was realized for a single-emitter laser device with an injection strip width of 190 μm, a ridge width of 300 μm, and a cavity length of 2 mm, with a final slope efficiency of 1.03 W/A, and with a horizontal divergence angle of about 13° and a vertical divergence angle of no more than 30°. The device has good slope efficiency, and this 100 ns pulse width can be effectively applied in the fields of fog-transparent imaging sensors and fast headroom ranging radar areas.

## 1. Introduction

The 1550 nm optical waveband, a commonly used atmospheric transmission window [1], has less attenuation in the atmosphere. Only a small portion of the light in this band reaches the human retina, and the safety threshold of the human eye is higher [2]. In addition, gas molecules, such as hydrogen sulfide and carbon monoxide, have a strong absorption for 1550 nm wavelength lasers [3]. Therefore, 1550 nm semiconductor lasers have a wide range of applications in the fields of LIDAR, fast-ranging long-distance optical communication, gas sensing, and so on. In the human eye safety band (>1300 nm), currently, the commonly used lasers are fiber lasers and semiconductor lasers. Fiber lasers have low electric–optical conversion efficiency, while semiconductor lasers not only have higher electric–optical conversion efficiency, but also have the advantages of small size, light weight, low power consumption, and high reliability. The current commercial 1.3 μm optical communication light sources are mainly InP-based InGaAsP or InGaAlAs quantum well (QW) lasers [4], while compared with the 1310 nm band, the 1550 nm band has a lower transmission loss, which is suitable for medium- and long-distance communication transmission [5]. Most of the existing 1550 nm semiconductor laser products are InP-based, but the energy band offset of the InP-based quantum well is small, so its temperature stability is not good. GaAs-based quantum well lasers have high temperature stability [6], but their wavelengths are limited to less than 1.2 μm due to bandgaps of GaAs-based materials [7]. Silicon-based light sources with the advantages of low cost and easy integration are the current research hotspots [8], but most of the existing semiconductor lasers are based on III-V compound materials, which are incompatible with the silicon-based complementary metal oxide semiconductor (CMOS) process, which seriously impedes the development of silicon-based light sources [9,10]. Although pure group IV semiconductor materials are compatible with the silicon-based CMOS process, group IV elemental semiconductors are indirect bandgap semiconductors, and the lasers developed have low luminous efficiencies. The growth of III-V materials on silicon substrates using wafer bonding and anisotropic epitaxy techniques [11,12] enables the creation of efficient and reliable laser light sources [13,14]. As far as the active region structure is concerned, the structure of 1550 nm LD is gradually evolving from quantum wells to 1550 nm quantum dots. Theoretically, quantum dot lasers are more tolerant of dislocations and have better stability [15], and the threshold current is less affected by temperature [16], but the current performance of quantum dot lasers is still far from the theoretical predictions [17] and also inferior to that of quantum well lasers.

Tunneling junction technology was first used for 905 nm semiconductor lasers [18,19], and then gradually for InP material lasers [20,21]. At present, most of the active regions of high-power 1550 nm semiconductor lasers adopt multiple quantum well structures, and connecting multiple quantum wells through electron tunneling can increase the output power of lasers and the optical field confinement factor, but the threshold current density will also increase [22]. Within the last 20 years, domestic and foreign countries have continuously improved the output power by optimizing the epitaxial structure and growth process to enhance the slope efficiency and reduce the tunnel junction resistance. In 2022, SemiNex reported a 1550 nm two-tunnel-junction high-power quantum well laser based on AlInGaAs/InP materials, with laser diodes connected by thin low-resistance tunnel junctions, a device ridge width of 350 μm, and a peak output power of more than 100 W at a final 100 A (pulse width of 10 ns), with a slope efficiency of 1 W/A [23]. In July 2024, the 13th Research Institute of China Electronics Technology Group Corporation reported a MOCVD-grown 1550 nm one-tunnel-junction laser, with the number of quantum wells and the doping concentration of the tunneling junction optimized, and reached an output power of about 18.5 W at a frequency of 1 KHz, a pulse width of 100 ns, and a pulsed current of 35 A at room temperature under the test conditions [24]. In this paper, we specially calculated the material composition of the whole laser epitaxial structure, to increase the output power and efficiency of the lasers. The epitaxial growth process was also optimized to obtain the best materials possible in terms of optical and electrical properties. The laser output spot was shaped through the optical lens system in order to improve the beam quality of the lasers. Finally, the laser P-I-V characteristics, the temperature drift characteristics of the output wavelength, and the beam divergence angle were tested.

## 2. Theoretical Analyses

The commonly used active region materials for 1550 nm semiconductor lasers are InGaAsP and InGaAlAs. InGaAlAs has many unique merits for high laser performance. It has a higher conduction band offset (ΔEc/ΔEg), which helps to improve the thermal escape carrier confinement, and reduces the carrier leakage in the quantum well region. The refractive index of InGaAlAs material is higher, which is good for providing good optical confinement [25]. The growth quality of InGaAlAs is better than that of InGaAsP, which is helpful in allowing it to realize good luminescence performance. The lattice constant of InGaAlAs is smaller than that of the InP substrate, which results in a tensile-strain effect, with a favorable energy band separation effect and high-level gain, so in this paper, InGaAlAs was chosen as the ideal quantum well layer material for 1550 nm semiconductor lasers [26].

### 2.1. Radiation Wavelength

The wavelength of a quantum well laser is determined by the effective bandgap Eeff in the active region [27], calculated as follows:(1)$$\lambda =\frac{\mathrm{hc}}{h\nu }\;\approx \;\frac{1.24}{E_{\mathrm{eff}}}$$
where λ is the laser wavelength, h is Planck’s constant, c is the propagation speed velocity of light in a vacuum, ν is the photon frequency, E_eff_ is the equivalent bandgap, and the units λ and E_eff_ are μm and eV, respectively. According to the quantum theory, E_eff_ is 0.8 eV for a λ value of 1.55 μm.

### 2.2. Quantum Well Compositions

A piezostrain quantum well layer and a tensile-strain barrier layer are used. The bandgap of the In_1−x−y_Ga_x_AlyAs material in the absence of strain is as follows [28]:(2)$$E_{g}=0.36+2.093y+0.629x+0.577y^{2}+0.436x^{2}+1.013\mathrm{yx}-2\left(1-x-y\right)\mathrm{xy}$$

All physical parameters of the In_1−x−y_Ga_x_AlyAs quaternary alloy except the bandgap can be obtained by linear interpolation:(3)$$P\left({\mathrm{In}}_{1-x-y}{\mathrm{Ga}}_{x}{\mathrm{Al}}_{y}\mathrm{As}\right)=\;\left(1-x-y\right)P\left(\mathrm{InAs}\right)+\mathrm{xP}\left(\mathrm{GaAs}\right)+\mathrm{yP}\left(\mathrm{AlAs}\right)$$

The following parameters of the In_1−x−y_Ga_x_Al_y_As quaternary alloy can be obtained using the data given in Table 1:(4)C11 =8.329+3.55x+4.171y
(5)$$\;C_{12}=4.526+0.85x+0.814y$$
(6)$$a_{e}=\;6.0584-0.4051x-0.3984y$$
(7)$$a_{c}=-5.08-2.09x-0.56y$$
(8)$$a_{\nu }=1+0.16x+1.47y$$
(9)b=−1.8+0.1x+0.3y

From a = a_c_ − a_ν_ and Equations (7) and (8), we have
(10)a=−6.08−2.25x−2.03y

The relationship between Eg(x,y) and material composition can be obtained from Equations (2)–(10), the relationship between In content and the In_1−x−y_Ga_x_Al_y_As material bandgap and strain can be obtained by simplification, and the material composition of the undoped QW layer is finally calculated to be Al_0.08_Ga_0.24_In_0.68_As, and that of the barrier layer is Al_0.13_Ga_0.43_In_0.44_As. The emission wavelength drifts with the width of the quantum well and temperature changes, and the wavelength needs to be correspondingly calibrated by a PL spectral tester, wavelength tester, etc., during the realization of the laser.

## 3. Device Structure

The device epitaxial structures were grown using a ThermoV90 gas source molecular beam epitaxy system (VG, East Sussex, UK), with high-purity solid-state sources of the group III elements In, Ga, and Al, and a group V source of As_2_ and P_2_ generated by the cracking of arsine (AsH_3_) and phosphine (PH_3_) at high temperatures (1040 °C). The n-type and p-type doping sources were high-purity Si and Be, respectively, and the substrate was chosen to be a Si-doped n-type (100) InP single-crystal wafer (1 × 10 ^18^ cm^−3^). The epitaxial structure of the device is shown in Table 2.

Three quantum wells connected by two tunnel junctions are used as the active region. The tunnel junction layer consists of 30 nm thick n-type 5 × 10^19^ heavily doped InP (n++) and heavily doped InAlAs materials: 5 nm thick In_0.22_Al_0.78_As (p++) and 10 nm thick In_0.22_Al_0.78_As (p+). Three undoped InGaAsAl quantum well structures were used in the active region, including a piezostrain quantum-well Al_0.08_Ga_0.24_In_0.68_As with a thickness of 7 nm and a tensile-strain barrier Al_0.13_Ga_0.43_In_0.44_As with a thickness of 20 nm. Above and below the active layer are the In_0.22_Al_0.78_As waveguide layer and InP cladding layer, and the highly doped p-In_0.22_Ga_0.78_As is used as the ohmic contact layer. The bandgaps of both InAlAs and InP are different, and the difference in the bandgaps between the two can limit the longitudinal light field and reduce carrier leakage, and the general diffusion length of the carriers is about 1 μm, so the limiting layer should not be too thick.

Compared with multiple lasers connected in series, the design of multiple active regions connected by a tunnel junction can avoid the problem of large distance between active regions caused by a thick substrate, which is conducive to reducing the divergence angle of the output beam. However, the coupling of the light field in each active region is critical. The structure of the tunnel junction is very close to that of an ordinary p-n junction or diode. The main difference is that the p-type layer and n-type layer forming the p-n junction are highly doped, and the doping concentration is ≥10^19^ cm^−3^, which is capable of generating a tunneling effect.

The energy band diagram of the tunnel junction is shown in Figure 1a, where an electron injected from the p-type electrode falls into the valence band radiative composite in the last active region and emits a photon, after which the electron falls into the valence band tunnels through the reverse-biased tunnel junction and emits another photon in the next active region [29]. In the highly doped p-type layer and n-type layer, the Fermi energy levels fall into the valence band and conduction band, respectively, and the junction region is narrow; due to the quantum tunneling effect, the n-region conduction band electrons may pass through the forbidden band to the p-region valence band, and the p-region valence band electrons may also pass through the forbidden band to the n-region conduction band, thus generating a tunneling current. When the reverse bias is added, the p-region energy band is elevated relative to the n-region energy band, and within the range of quantum states with the same energy on both sides of the junction, the quantum states below the Fermi energy level in the valence band of the p-region are occupied by the electrons, while the n-region conduction band has empty quantum states above the Fermi energy level, the potential barrier region is thinner than that at equilibrium, the tunneling length decreases, and the probability of electrons in the valence band in the p-region passing through the tunneling is increased with the increase in the reverse bias voltage; therefore, the reverse current also increases rapidly. Therefore, the tunnel junction has good conductivity in reverse bias mode. The I–V test curve and reverse bias resistance of the tunnel junction at 25 °C are shown in Figure 1b. It can be seen that the forward turn-on voltage of the p+-InAlAs-n+-InP tunnel junction is less than 0.1 V. Under reverse bias, it has a large reverse current, and the slope of the curve is basically equal to that in the forward direction, and the forward negative resistance characteristic is not observed. The bias resistance of the tunnel junction in reverse operation is about 22 Ω.

The device’s cross-section structure is shown in Figure 2a. Since most of the optical loss occurs in the sidewalls of the ridge, in this paper, we adopted a ridge-shaped wide waveguide structure to limit the transverse optical field and reduce the thermal resistance, and to reduce the effects of the optical loss and the waste heat to enhance the laser performance, so as to achieve the purpose of lowering the thermal power density and the beam divergence angle. In order to reduce the transverse expansion current and improve the power output, two trenches with a depth exceeding the tunnel junction were photolithographed perpendicular to the growth direction of the epitaxial layer, and the trenches were filled with SiO_2_. The electrode window was formed by photolithography. The upper electrode was sputtered with Ti/Pt/Au, and the backside was thinned and evaporated with AuGe/Ni/Au. A single-emitter device was obtained by cleaving, and sintered into a flip-flop structure, press welded, and packaged into the final device. The structural appearance of the single-emitter laser device is shown in Figure 2b, with an injection strip width of 190 μm, a ridge width of 300 μm, and a cavity length of 2 mm. An SEM image of the exit surface of the finally fabricated 1550 nm tunnel junction QW laser is shown in Figure 2c.

## 4. Results and Discussion

The output power, lasing wavelength, spectral width, divergence angle, and temperature characteristics of the device were tested to determine whether the design objectives were met. The laser chip was operated under pulsed condition, and the output power, emission spectrum, and temperature characteristics of the device were tested using a Centauri Dual Channel optical energy meter equipped with a PE9-ES-C probe (Ophir Optronics, Jerusalem, Israel) and an HR4000CG-UV-NIR spectral analyzer (Ocean Optics, Orlando, FL, USA).

### 4.1. P-I-V, Spectrogram, and Temperature Drift Coefficient

The actual measured P-I-V characteristics of the device, when the cavity surface was not coated, are shown in Figure 3a. The threshold current of the 1550 nm single-emitter laser is ~500 mA, and the turn-on voltage is 2.05 V. At a drive current of 30 A, the peak output power reaches 31 W at room temperature (repetition frequency 10 kHz and pulse width 100 ns), and the slope efficiency is 1.03 W/A. Our metrics are comparable with Ref. [23] in terms of the peak pulse power achieved at 30 A, but the slope efficiency is slightly better. In addition, we tested the laser performance at a pulse repetition frequency of 10 kHz and a pulse width of 100 ns, compared to their 5 ns and 150 ns pulse widths, in view of the practical application demand, where high peak power lasers at this particular pulse parameter are required for fog-transparent imaging sensors and fast headroom ranging radars. Figure 3b shows the measured wavelength values of the laser at different temperatures. The central wavelength is 1551.1 nm at 20 °C, and the wavelength red-shifted from 1545 nm to 1560 nm for the temperature range of 10 °C to 35 °C, with a temperature drift coefficient of about 0.6 nm/°C. Figure 3c shows the spectrum obtained from the actual measurements at 30 A, 25 °C.

### 4.2. Testing of Far-Field Spot

We obtained the laser spot at the light-emitting surface of the device under a metallurgical microscope with the short-wave color camera model ARTCAM-990SWIR (Artray, Tokyo, Japan), as shown in Figure 4a, which clearly shows three luminescent spots corresponding to the three quantum wells in the laser structure. Beam shaping of the flip-coated device was carried out using an optical lens, and the far-field spot was tested with a short-wave color camera; Figure 4b shows the construction scheme for the test equipment.

We placed the receiving screen at about 50 cm from the luminous surface. Figure 5a shows the far-field pattern taken by the camera without collimation; the test revealed the existence of three luminous layers with different intensity distributions, and the first stripe in the figure is near the substrate. There are obvious dark lines between the first and second stripes, the second and third stripes are close to the P-side surface, where the stripe is strongest, and there are no dark lines between the second and third stripes, which almost overlap. A 200 μm diameter microlens was used to compress and collimate the light spots along the fast-axis direction to obtain a nearly square light spot as shown in Figure 5b. Figure 5c shows the far-field distribution of the final resulting spot, with a vertical divergence angle close to 30° and a horizontal divergence angle of about 13°.

## 5. Conclusions

In this paper, the emitting wavelength of a laser and the material components of the active region of a quantum well were designed through theoretical calculations, and the core of a dual-channel ridge waveguide structure with an injection strip width of 190 μm and a cavity length of 2 mm was prepared with an MBE epitaxial growth technology and chip process, and the pulsed output power of the tunneling junction laser with three active regions of the quantum wells at room temperature reached 31 W (10 KHz, 100 ns), with a slope efficiency of 1.03 W/A. The lasing wavelength red-shifted from 1545 nm to 1560 nm over the temperature range of 10 °C to 35 °C, with a temperature drift coefficient of about 0.6 nm/°C. The far-field test of the laser with a short-wave infrared color camera shows that there are three outgoing spots, which is consistent with the theoretical design of the three quantum-well active regions, and the device is optically beam shaped to obtain a near-square laser spot, with a vertical divergence angle of close to 30°, and a horizontal divergence angle of about 13°, which can allow it to meet the application requirements of 1550 nm semiconductor lasers for a wide range of occasions. The 1550 nm human eye-safe high-power tunnel junction semiconductor laser with low attenuation in air not only improves the detection distance and resolution of LIDAR, but also enhances the communication transmission distance, and is expected to generally replace 905 nm LDs in the future in the fields of LIDAR, precise ranging, and laser indication. In particular, the peak power of the pulse we obtained at a 100 ns pulse width can be better applied in the fields of fog-transparent imaging sensors and fast headroom ranging radars.

## Figures and Tables

**Figure 1 micromachines-15-01042-f001:**
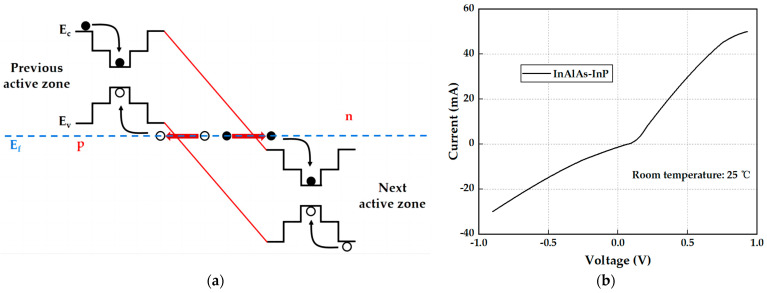
InAlAs-InP tunnel junction: (**a**) energy band diagram of tunnel junction (The black lines represent the E_c_ and E_v_, the blue line represents the E_f_, and the red lines represent the tunnel junction region.); (**b**) I–V curve of tunnel junctions at room temperature.

**Figure 2 micromachines-15-01042-f002:**
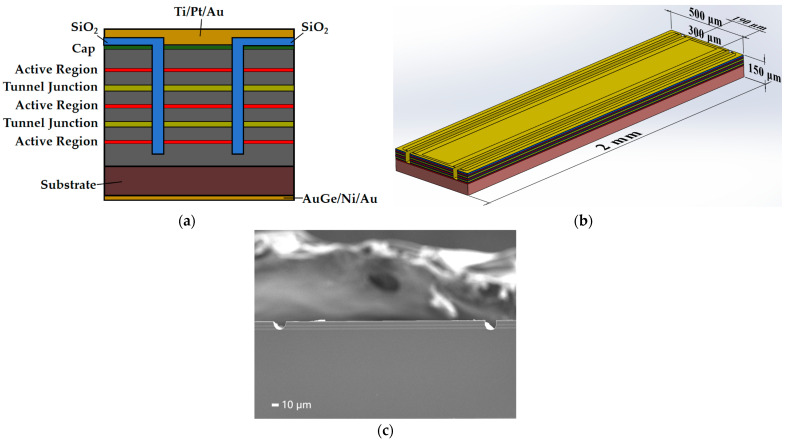
Structure diagram of 1550 nm semiconductor laser: (**a**) cross-section of a dual-channel structured tunnel junction semiconductor laser chip; (**b**) single-emitter laser device structure; (**c**) SEM image of the exit surface of the device.

**Figure 3 micromachines-15-01042-f003:**
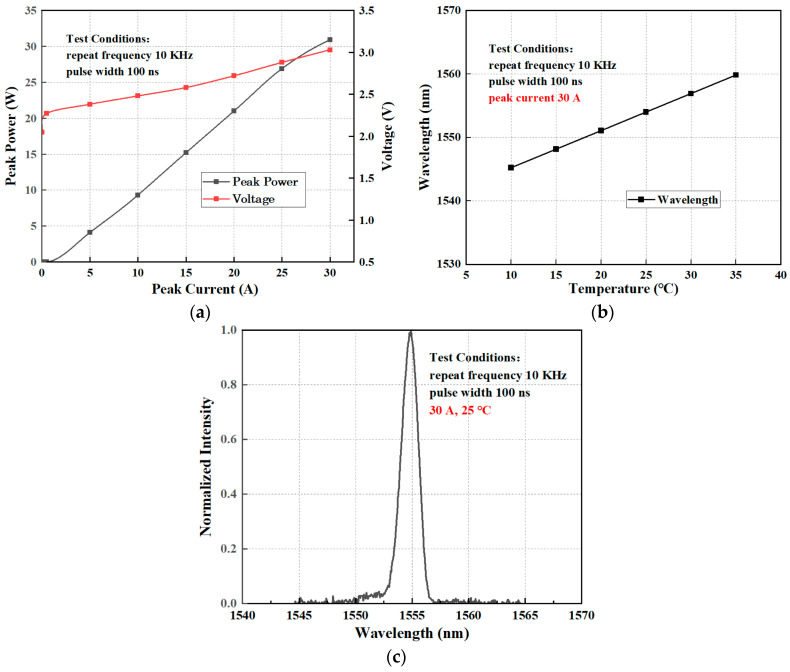
Test results for 1550 nm eye-safe pulse semiconductor laser: (**a**) P-I-V diagram; (**b**) a wavelength shift caused by temperature changes; (**c**) light emission spectrum.

**Figure 4 micromachines-15-01042-f004:**
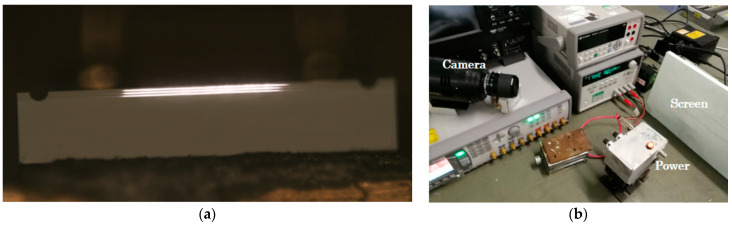
Spot test for 1550 nm laser. (**a**) The laser spot at the luminescent surface of the device under a metallurgical microscope; (**b**) devices for testing laser spots.

**Figure 5 micromachines-15-01042-f005:**
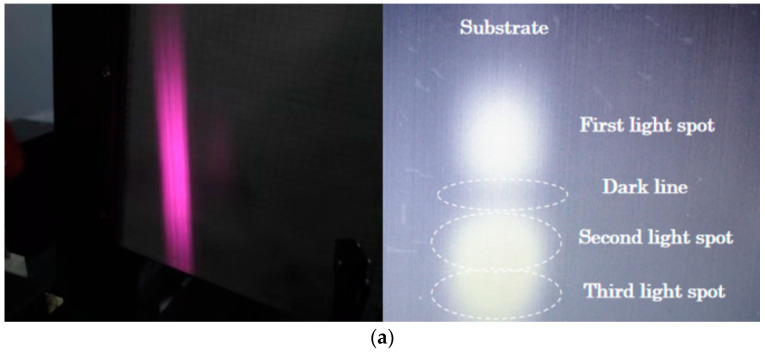
Far-field pattern for 1550 nm laser. (**a**) Uncollimated; (**b**) compression along the fast-axis direction with a 200 μm diameter microlens; (**c**) far-field divergence angles of a 1550 nm laser at 10 W.

**Table 1 micromachines-15-01042-t001:** Physical parameters of In_1−x−y_Ga_x_Al_y_As-related binary semiconductor compounds [27].

Parameters	Unit	AlAs	GaAs	InAs	InP
Lattice constant	a (Å)	5.660	5.6533	6.0584	5.8688
Elastic hardness constant	C_11_ (10^11^ dyn/cm^2^)	12.5	11.879	8.329	10.11
Elastic hardness constant	C_12_ (10^11^ dyn/cm^2^)	5.34	5.376	4.526	5.61
Static distortion potential of the conduction band	a_c_ (eV)	−5.64	−7.17	−5.08	−5.04
Valence band static distortion potential	a_v_ (eV)	2.47	1.16	1.00	1.27
Valence band tangent distortion potential	b (eV)	−1.5	−1.7	−1.8	−1.7
Parameters of valence band	γ _1_	3.45	6.8	20.4	4.95
	γ _2_	0.68	1.9	8.3	1.65
	γ _3_	1.29	2.73	9.1	2.35
Electronic effective mass	m_e_/m_o_	0.15	0.067	0.023	0.077
Heavy hole effective mass	m_hh_/m_o_	0.79	0.50	0.40	0.60
Light hole effective mass	m_lh_/m_o_	0.15	0.076	0.025	0.12

**Table 2 micromachines-15-01042-t002:** Epitaxial structure of 1550 nm semiconductor laser.

Number	Layers	Materials	Thickness (μm)	Doping (cm^−3^)
20	p+-Contact	In_0.22_Ga_0.78_As	50 nm	Be, 2 × 10^19^
19	p-Smooth	In_0.22_Ga_0.78_As_0.92_P_0.08_	0.2	Be, 5 × 10^18^
18	p-Cladding	InP	1.2	Be, 5 × 10^18^
17	Waveguiding	In_0.22_Al_0.78_As	0.25	undoped
16	QW	Al_0.08_Ga_0.24_In_0.68_As/ Al_0.13_Ga_0.43_In_0.44_As	7 nm/ 20 nm	undoped
15	Waveguiding	In_0.22_Al_0.78_As	0.25	undoped
14	n-Cladding	InP	1.2	Si, 1 × 10^18^
13	Tunnel junction	InP (n++)	10 nm	Si, 5 × 10^19^
		In_0.22_Al_0.78_As (p++)	5 nm	Be, 1 × 10^20^
		In_0.22_Al_0.78_As (p+)	10 nm	Be, 1 × 10^19^
12	p-Cladding	InP	1.2	Be, 5 × 10^17^
11	Waveguiding	In_0.22_Al_0.78_As	0.25	undoped
10	QW	Al_0.08_Ga_0.24_In_0.68_As/Al_0.13_Ga_0.43_In_0.44_As	7 nm/20 nm	undoped
9	Waveguiding	In_0.22_Al_0.78_As	0.25	undoped
8	n-Cladding	InP	1.2	Si, 1 × 10^18^
7	Tunnel junction	InP (n++)	10 nm	Si, 5 × 10^19^
		In_0.22_Al_0.78_As (p++)	5 nm	Be, 1 × 10^20^
		In_0.22_Al_0.78_As (p+)	10 nm	Be, 1 × 10^19^
6	Etching stop	In_0.22_Ga_0.78_As_0.92_P_0.08_	0.2	Be, 5 × 10^17^
5	p-Cladding	InP	1.2	Be, 5 × 10^17^
4	Waveguiding	In_0.22_Al_0.78_As	0.25	undoped
3	QW	Al_0.08_Ga_0.24_In_0.68_As/Al_0.13_Ga_0.43_In_0.44_As	7 nm/20 nm	undoped
2	Waveguiding	In_0.22_Al_0.78_As	0.25	undoped
1	n-Cladding	InP	0.5	Si, 1 × 10^18^
	Substrate	InP	300	Si, 1 × 10^18^

## Data Availability

The original contributions presented in this study are included in this article. Further inquiries can be directed to the corresponding author.

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
