# Peer review of "Study on 1550 nm Human Eye-Safe High-Power Tunnel Junction Quantum Well Laser"

_micromachines, 2024, doi:10.3390/mi15081042_

Round 1

Reviewer 1 Report

Comments and Suggestions for Authors

The topic of the article is very relevant. both from fundamental and applied points of view. To improve the presentation of the material, in my opinion, it is necessary to make a number of improvements. In the abstract of the article, it is necessary to more specifically highlight the main result (results) obtained for the first time by the authors and significantly different from what was obtained by other scientific teams. In the introduction, it is necessary to describe in more detail the state of affairs in this scientific direction and focus on what has not been done before and why, but has been implemented in this work. In terms of theoretical analysis, it is necessary to describe the formulas used in more detail and provide appropriate references (formula 1). When using numerical calculations and expressions with constants, it is necessary to provide references from where these constants were taken (2).

            On the experimental graphs shown, it is necessary to indicate the experimental errors characterizing the accuracy of the measurements. The quality of some of the pictures needs to be improved, for example, Figure 3c needs to be presented in graph form. Also, Figure 4 should be significantly improved and presented with figures characterizing the spatial distribution

            In conclusion, it is necessary to show in more detail the novelty of the results obtained, especially in terms of the connection between theoretical estimates and the results obtained.

Author Response

Thank you for your comments and questions. We have responded to your comments. Please see the attachment.

Reviewer 2 Report

Comments and Suggestions for Authors

Micromachines MDPI

Study on 1550 nm Human Eye-safe High-power Tunnel Junction Quantum Well Laser

Manuscript Number: micromachines-3134470

Article Type: Full-Length Article

Section/Category: Lasers, optics and photonics

Reviewer observation

The research manuscript is significant because the authors present theoretical results for building and designing a “High-Power Tunnel Junction Quantum Well Laser” relevant results.

1.     The paper is a good proposal and shows exciting results supported by theoretical calculations; I only have a few Observations/Comments….

2.     The Introduction section needs to include actualized references, and the explanation of this section must be adequate and concordant. The References only have two from 2018 to 2020; the rest are ten or older.  

3.     In concordance with the last point, there are other developments with a bigger pulsed peak power (>100W) than the report in the manuscript in the literature; please justify why it is appropriate to publish your results. Sorry for not including references; you should look for current references.

4.     In section 4.1, report “repetition frequency 10 kHz and pulse width 100 ns,” Could you explain why the result is essential? Indeed, there are again other published papers with pulse width <10ns.

5.     The results are acceptable, but they need to be improved.

Author Response

(The authors gave the same response as above.)

Reviewer 3 Report

Comments and Suggestions for Authors

In this manuscript, a multi-junction 1550nm EEL with high pulse power was reported, which has potential applications of lidar and sensing. The topic is of interest to the community. However, a couple of points need to be addressed to make the content even clearer. Please refer to the details listed below:

1. The model of material physical parameters is not precise. The empirical formula for the calculation of In1-x-yGaxAlyAs physical parameters should include appropriate bowing parameters, which have been ignored in this paper.

2. The laser epitaxial structure utilized 250nm Al0.22In0.78As as n- and p-waveguide layers. However, Al0.22In0.78As exhibits significant strain on the InP substrate. Has the critical thickness been taken into consideration?

3. On page 5, line 132, a multi-junction cascaded laser may not necessarily be beneficial for reducing the divergence angle unless there is coupling of the light field from each active region.

4. Please carefully review the voltage-current curve in Fig. 3(a). For a multi-junction diode laser, the turn-on voltage usually doubles with the number of junctions. Fig 3(c) is unclear and requires clarification. Furthermore, it needs to be explained why the slope efficiency increases from 25A to 30A in Fig. 3(a).

5. The spectral width is large, and there is side peaks. Is the peak wavelength or central wavelength more suitable for calculating the temperature shift coefficient of wavelength?

Comments on the Quality of English Language

The English language needs to be reviewed and properly edited for plurality, grammar, and meaning. Additionally, there are many grammatical and terminology errors in the manuscript; for example, it should be "Valence band" instead of price band, and hole instead of vacua.

Author Response

(The authors gave the same response as above.)

Round 2

Reviewer 2 Report

Comments and Suggestions for Authors

Micromachines MDPI,

Study on 1550 nm Human Eye-safe High-power Tunnel Junction Quantum Well Laser

Manuscript Number: micromachines-3134470

Article Type: Full-Length Article

Section/Category: Lasers, optics and photonics

Comments: I reviewed the new version of the manuscript “Study on 1550 nm Human Eye-safe High-power Tunnel Junction Quantum Well Laser,” which has been improved based on my observations. I don’t have any more comments; the manuscript could be published in its last version.

Reviewer 3 Report

Comments and Suggestions for Authors

The authors have  revised the original manuscript in accordance with the recommendations of the reviewers. I suggest that this paper be considered for publication in Micromachines. However, it is advisable for the authors to thoroughly review the following papers ([1] Physica E, 5:215 (2000); [2] J. Appl. Phys. 134:243103 (2023)).